# REVISITING BEHAVIOR REGULARIZED ACTOR-CRITIC

**Denis Tarasov, Vladislav Kurenkov, Alexander Nikulin, Sergey Kolesnikov**
Tinkoff AI
`{den.tarasov, v.kurenkov, a.p.nikulin, s.s.kolesnikov}@tinkoff.ai`

## ABSTRACT

In recent years, significant advancements have been made in offline reinforcement learning, with a growing number of novel algorithms of varying degrees of complexity. Despite this progress, the significance of specific design choices and the application of common deep learning techniques remains unexplored. In this work, we demonstrate that it is possible to achieve state-of-the-art performance on the D4RL benchmark through a simple set of modifications to the minimalist offline RL approach and careful hyperparameter search. Furthermore, our ablations emphasize the importance of minor design choices and hyperparameter tuning while highlighting the untapped potential of using deep learning techniques in offline reinforcement learning.

## 1    INTRODUCTION

Offline reinforcement learning has seen remarkable progress in recent years, particularly in the domains of robotics (Smith et al., 2022; Kumar et al., 2021) and recommender systems (Chen et al., 2022a). The vast number of potential applications and tasks has led to the frequent release of new offline RL algorithms, as reported in various studies (Levine et al., 2020; Prudencio et al., 2022). This rapid pace of development is both exciting and challenging, as it shows the need for novel approaches but also requires a careful evaluation of the seemingly minor design choices made in the introduced algorithms.

To be more precise, there are at least three branches of research that are relevant to this question. First, the usage of deeper networks, which were initially introduced in CQL (Kumar et al., 2020), has gained widespread acceptance and has been implemented in various other offline RL techniques (An et al., 2021; Yang et al., 2022; Zhuang et al., 2023). Second, Layer Normalization, which was initially used in online RL studies (Hiraoka et al., 2021), has now become a common practice in offline settings as well (Nikulin et al., 2022; Kumar et al., 2022; Ball et al., 2023). Third, recent advancements in ensemble-based methods (An et al., 2021; Yang et al., 2022; Ghasemipour et al., 2022) have given rise to the development of large-batch offline RL, which has yet to be applied beyond the SAC-N method (An et al., 2021; Nikulin et al., 2022). These design choices become even more intricate with the advent of more advanced transformer-based offline RL methods (Chen et al., 2021; Janner et al., 2021), which bring together deeper networks, layer normalization, and other modifications at the same time. Overall, it remains an open question whether these small improvements represent only a minor gain in performance, or if they hold the key to the success of new algorithms.

In order to answer this question, we re-assess the impact of these ostensibly minor design choices by taking a simpler approach. Our findings demonstrate that by incorporating a certain set of these into the BRAC algorithm (Wu et al., 2019), and conducting a thorough hyperparameter search, it is possible to attain performance that rivals or surpasses state-of-the-art results on the AntMaze and locomotion tasks as evaluated by the D4RL benchmark. Our experiments utilize an ensemble-free TD3 algorithm (Fujimoto et al., 2018), but the principles and techniques can be easily extended to other actor-critic RL methods.

## 2 PRELIMINARIES

### 2.1 OFFLINE REINFORCEMENT LEARNING

Reinforcement Learning problem is usually defined as a Markov Decision Process (MDP) with the tuple $\{S, A, P, R, \gamma\}$, where $S \subset \mathbb{R}^n$ is a state space, $A \subset \mathbb{R}^m$ is an action space, $P : S \times A \to S$ is a transition function, $R : S \times A \to \mathbb{R}$ is a reward function, and $\gamma \in (0, 1)$ is a discount factor. The ultimate objective is to find a policy $\pi(a|s)$ that maximizes a cumulative discounted return $\mathbb{E}_\pi \sum_{t=0}^{\infty} \gamma^t R(s_t, a_t)$. Policy is supposed to improve while interacting with the environment by observing states and committing actions which provide some rewards.

Policies in the offline RL mode are not permitted to interact with the environment and can only access the static transactions dataset $D$ collected by one or more other policies. Such a setting brings new challenges, for example, the estimation of values for state-action pairs that are not presented in the dataset while exploration is not available (Levine et al., 2020).

### 2.2 BEHAVIOR REGULARIZED ACTOR-CRITIC

Behavior Regularized Actor-Critic (BRAC) is an offline RL framework introduced in Wu et al. (2019). The core idea behind BRAC is that actor-critic algorithms can be penalized in two ways to solve offline RL tasks: actor penalization and critic penalization. In this framework, the actor objective is represented as in Equation 1, and the critic objective as in Equation 2, where $F$ is a divergence function between dataset actions and policy actions distributions. The differences from a vanilla actor-critic are highlighted in blue.

$$\pi = \arg\max_\pi \mathbb{E}_{(s,a)\sim D}\left[Q_\theta(s, \pi(s)) - \alpha \cdot F(\pi(s), a)\right] \tag{1}$$

$$\theta = \arg\min_\theta \mathbb{E}_{\substack{(s,a,r,s',\hat{a}')\sim D \\ a'\sim\pi(s')}}\left[(Q_\theta(s, a) - (r + \gamma(Q_{\bar{\theta}}(s', a') - \alpha \cdot F(a', \hat{a}'))))^2\right] \tag{2}$$

In the original work, various choices of $F$ were evaluated when used as the regularization term for the actor or critic. The authors tested KL divergence, Kernel MMD, and Wasserstein distance but did not observe any consistent advantage. Finally, it is important to note that, originally, only one of the regularizations was enabled during experiments, resulting in only one of the $\alpha$'s being non-zero.

Subsequently, TD3 + BC (Fujimoto & Gu, 2021) was introduced, utilizing Mean Squared Error (MSE) as the regularization term $F$ for the actor.

## 3 REBRAC: A RECIPE

Our proposed method is a modified version of the BRAC algorithm, built upon the TD3 method (Fujimoto et al., 2018). There are *three* key differences from the original BRAC approach. *First*, we use mean squared error as the measure of divergence between the actor and dataset action distributions, which we found to be both simple and effective. *Second*, we separate the hyperparameter that governs the penalty strength into two separate components, one for the actor and one for the critic. This decoupling of penalties was shown to be useful in previous work (Rezaeifar et al., 2022). We refer to our approach as **Re**visited **BRAC**. The objective for the actor is outlined in Equation 3, and that for the critic is presented in Equation 4.

$$\pi = \arg\max_\pi \mathbb{E}_{(s,a)\sim D}\left[Q_\theta(s, \pi(s)) - \beta_1 \cdot (\pi(s) - a)^2\right] \tag{3}$$

$$\theta = \arg\min_\theta \mathbb{E}_{\substack{(s,a,r,s',\hat{a}')\sim D \\ a'\sim\pi(s')}}\left[(Q_\theta(s, a) - (r + \gamma(Q_{\bar{\theta}}(s', a') - \beta_2 \cdot (a' - \hat{a}')^2)))^2\right] \tag{4}$$

We also adopt the normalization of the Q function when calculating the actor loss, following the methodology outlined in Fujimoto & Gu (2021). This modification enhances the algorithm's robustness to the choice of the $\beta_1$ hyperparameter across varying environments. It is worth noting that when $\beta_2 = 0$, our approach is equivalent to TD3 + BC.

*Finally*, we **re**visited commonly utilized techniques in both deep learning, reinforcement learning, and offline reinforcement learning algorithms. To the best of our knowledge, there has been no extensive analysis of the impact of these techniques on proposed methods in offline RL to date.

**Deeper Networks.** Some of the previous offline RL algorithms, such as BRAC and TD3 + BC, utilized three-layer networks for both actor and critic. Recent research in the field, however, has shown that using four-layer networks, as seen in Kumar et al. (2020); An et al. (2021); Yang et al. (2022), can lead to better results. As such, we have taken the approach of increasing the depth of the actor and critic networks to four layers each in our study.

**Layer Normalization.** Layer Normalization, or LayerNorm (Ba et al., 2016), is a well-established technique in deep learning and online RL (Hiraoka et al., 2021). The application of normalization generally results in faster convergence and more stable training processes. While some promising results in offline RL have been obtained (Nikulin et al., 2022; Kumar et al., 2022; Ball et al., 2023), its application is still limited. In our work, we integrate LayerNorm into the critic networks by adding normalization between each layer.

**Large Batches.** Large batch optimization, a technique commonly used in deep learning, is another procedure we examine in our work. This approach typically results in faster convergence (You et al., 2017; 2019). In line with the findings in Nikulin et al. (2022), we adjust the batch size and increase the learning rate to improve convergence in offline RL tasks. Specifically, we increase the batch size from the default value of 256 to 1024.

These modifications are straightforward to implement, requiring only a few lines of code, and do not result in significant computational overhead. Despite being expected to produce only a minor performance improvement, we will further demonstrate that they have a significant impact on the final result in Section 4.2.

# 4 EXPERIMENTS

## 4.1 EVALUATION ON D4RL

Table 1: ReBRAC evaluation on the locomotion tasks. We report final normalized score averaged over 4 random seeds on v2 datasets. We highlight best scores over all algorithms with **bold** and best scores among ensemble-free algorithms with red. TD3 + BC and IQL scores are taken from Lyu et al. (2022). CQL, SAC-N and EDAC scores are taken from An et al. (2021). BRAC scores are taken from Chen et al. (2021). CNF scores are taken from Akimov et al. (2022). BPPO scores are taken from Zhuang et al. (2023). RORL scores are taken from Yang et al. (2022).

| | Ensemble-based | | | Ensemble-free | | | | | | |
|---|---|---|---|---|---|---|---|---|---|---|
| Task | SAC-N | EDAC | RORL | BRAC-v | TD3+BC | IQL | CQL | CNF | BPPO | ReBRAC, our |
| halfcheetah-medium | **67.5** ± 1.2 | **65.9** ± 0.6 | **66.8** ± 0.7 | 46.3 | 48.3 ± 0.3 | 47.4 ± 0.2 | 46.9 ± 0.4 | 50.5 ± 0.5 | 44.0 ± 0.2 | **65.9** ± 1.1 |
| halfcheetah-medium-expert | **107.1** ± 2.0 | **106.3** ± 1.9 | **107.8** ± 1.1 | 41.9 | 90.7 ± 4.3 | 86.7 ± 5.3 | 95.0 ± 1.4 | 96.2 ± 0.2 | 92.5±1.9 | **105.6** ± 1.4 |
| halfcheetah-medium-replay | **63.9** ± 0.8 | **61.3** ± 1.9 | **61.9** ± 1.5 | 47.7 | 44.6 ± 0.5 | 44.2 ± 1.2 | 45.84 ± 0.31 | 45.3 ± 0.3 | 41.0 ± 0.6 | 52.2 ± 4.6 |
| hopper-medium | 100.3 ± 0.3 | 101.6 ± 0.6 | **104.8** ± 0.1 | 31.1 | 59.3 ± 4.2 | 66.2 ± 5.7 | 61.9 ± 6.4 | 69.3 ± 1.0 | 93.9±3.9 | 102.5 ± 0.3 |
| hopper-medium-expert | 110.1 ± 0.3 | 110.7 ± 0.1 | **112.7** ± 0.2 | 0.8 | 98.0 ± 9.4 | 91.5 ± 14.3 | **96.9** ± 15.1 | **108.6** ± 5.4 | **112.8**±1.7 | 110.6 ± 2.0 |
| hopper-medium-replay | **101.8** ± 0.5 | 101.0 ± 0.5 | **102.8** ± 0.5 | 0.6 | 60.9 ± 18.8 | **94.7** ± 8.6 | 86.3 ± 7.3 | 89.0 ± 10.3 | 92.5 ± 3.4 | 99.9 ± 1.1 |
| walker2d-medium | 87.9 ± 0.2 | 92.5 ± 0.8 | **102.4** ± 1.4 | 81.1 | 83.7 ± 2.1 | 78.3 ± 8.7 | 79.5 ± 3.2 | 83.6 ± 3.0 | 83.6 ± 0.9 | 86.1 ± 0.2 |
| walker2d-medium-expert | 116.7 ± 0.4 | 114.7 ± 0.9 | **121.2** ± 1.5 | 81.6 | 110.1 ± 0.5 | 109.6 ± 1.0 | 109.1 ± 0.2 | 112.3 ± 0.2 | 113.1 ± 2.4 | 111.9 ± 0.4 |
| walker2d-medium-replay | 78.7 ± 0.7 | 87.1 ± 2.4 | **90.4** ± 0.5 | 0.9 | 81.8 ± 5.5 | 73.8 ± 7.1 | 76.8 ± 10.0 | 81.9 ± 1.98 | 77.6 ± 7.8 | **83.2** ± 10.3 |
| Average | 92.6 | 93.4 | **96.7** | 36.8 | 75.2 | 76.9 | 77.5 | 81.8 | 83.4 | 90.9 |

We evaluate the proposed approach on two commonly used D4RL benchmarks, the locomotion and AntMaze tasks. For the locomotion tasks, we select medium, medium-replay, and medium-expert datasets for HalfCheetah, Hopper, and Walker2d environments. For AntMaze, we consider all of the available datasets. We compare our results to several ensemble-free baselines, including TD3 + BC (Fujimoto & Gu, 2021), IQL (Kostrikov et al., 2021), CQL (Kumar et al., 2020), and BPPO (Zhuang et al., 2023), as well as to an ensemble-based baseline, RORL (Yang et al., 2022). BRAC (Wu et al., 2019) and CNF (Akimov et al., 2022) scores are reported for the locomotion tasks[1].We also compare

---

[1]BRAC and CNF scores are not available for AntMaze tasks.

Table 2: ReBRAC evaluation on AntMaze tasks. We report the final normalized score averaged over 4 random seeds on v1 datasets. We highlight the best scores overall algorithms with **bold** and the best scores among ensemble-free algorithms with red. IQL, CQL, and MSG scores are taken from Ghasemipour et al. (2022). TD3+BC, RORL scores are taken from Yang et al. (2022).

| Task Name | Ensemble-based | | Ensemble-free | | | | |
| --- | --- | --- | --- | --- | --- | --- | --- |
| | RORL | MSG | TD3+BC | IQL | CQL | BPPO | ReBRAC, our |
| antmaze-umaze | $97.7 \pm 1.9$ | $97.8 \pm 1.2$ | 78.6 | 87.5 | 74.0 | $95.0 \pm 5.5$ | $99.2 \pm 0.9$ |
| antmaze-umaze-diverse | $90.7 \pm 2.9$ | $81.8 \pm 3.0$ | 71.4 | 62.2 | 84.0 | $91.7 \pm 4.1$ | $98.0 \pm 2.1$ |
| antmaze-medium-play | $76.3 \pm 2.5$ | $89.6 \pm 2.2$ | 10.6 | 71.2 | 61.2 | $51.7 \pm 7.5$ | $84.5 \pm 4.7$ |
| antmaze-medium-diverse | $69.3 \pm 3.3$ | $88.6 \pm 2.6$ | 3.0 | 70.0 | 53.7 | $70.0 \pm 6.3$ | $82.5 \pm 7.5$ |
| antmaze-large-play | $16.3 \pm 11.1$ | $72.6 \pm 7.0$ | 0.2 | 39.6 | 15.8 | $86.7 \pm 8.2$ | $88.2 \pm 3.7$ |
| antmaze-large-diverse | $41.0 \pm 10.7$ | $71.4 \pm 12.2$ | 0.0 | 47.5 | 14.9 | $88.3 \pm 4.1$ | $90.7 \pm 4.8$ |
| Average | 65.2 | 83.6 | 27.3 | 63.0 | 50.6 | 80.5 | **90.5** |

to ensemble-based methods, SAC-N/EDAC (An et al., 2021) for the locomotion tasks[2] and MSG (Ghasemipour et al., 2022) for AntMaze tasks[3].

The majority of the hyperparameters are adopted from TD3 + BC, and $\beta_1$ and $\beta_2$ parameters from Equations 3 and 4 are carefully tuned. We examine the sensitivity to these parameters of the proposed approach in Appendix C. For a complete overview of the experimental setup and details, see Appendix A.

The results of our tests on the D4RL locomotion and AntMaze benchmarks are available in Table 1 and Table 2, respectively. Our approach, ReBRAC, shows a notable improvement in performance compared to the ensemble-free algorithms on the locomotion datasets. Moreover, on AntMaze tasks, ReBRAC achieves state-of-the-art results, outperforming both ensemble-free and ensemble-based algorithms on average. An important finding in our experiments was the need to change the $\gamma$ value from the default of 0.99 to 0.999 for all AntMaze tasks, as well as the careful tuning of learning rate parameters for the actor and critic networks. These adjustments contributed to the improved performance of ReBRAC, see Appendix A for more details.

## 4.2 ABLATIONS

To better understand the source of improved performance, we conducted an ablation study on the modifications made to the algorithm. Results can be found in Table 3. Additional ablation studies for all datasets can be found in Appendix D. One modification at a time was disabled, while all other modifications were retained, including: layer normalization in the critic network, additional linear layers in the actor and critic networks, adding an MSE penalty to the critic and actor loss, and large batches usage. In the case of AntMaze, we also attempted to use the default $\gamma$ value instead of the increased one. To further demonstrate the efficacy of our modifications, we also ran our implementation as equivalent to the original TD3 + BC, with all modifications disabled and hyperparameters taken from the original paper. This serves to show that the improved scores are due to the proposed changes in the algorithm and not just different implementations. Furthermore, we performed a search for the regularization parameter for our TD3 + BC to show that tuning this parameter is not the sole source of improvement.

As a result of our experiments, we can draw several conclusions. First, the improvement in performance is not due to differences in the implementation, as shown by the results of "TD3 + BC, our". Second, while tuning the regularization parameter in TD3 + BC results in better performance, it still falls short of the results achieved by ReBRAC, as demonstrated by "TD3 + BC, tuned". Our findings also suggest that changes to the $\gamma$ value have a significant impact on the ability to solve AntMaze tasks. Furthermore, disabling layer normalization results in poor scores that are worse than disabling any other modification except actor penalty, resulting in performance worse than "TD3 + BC, tuned". Similarly, removing additional layers lowers performance to that of tuned TD3 + BC. The actor penalty plays a crucial role, and its absence results in a noticeable drop in performance. Disabling both the critic penalty or large batches also has a negative impact on performance, albeit to a lesser

---

[2]SAC-N and EDAC score 0 on medium and large AntMaze tasks (Tarasov et al., 2022).

[3]MSG numerical results are not available for locomotion tasks.

Table 3: ReBRAC ablations results averaged over domains. Each modification was disabled for ReBRAC while keeping all the other.

| Ablation | Locomotion | AntMaze | All |
|---|---|---|---|
| TD3 + BC, paper | 75.2 | 27.3 | 56.0 |
| TD3 + BC, our | 62.9 | 22.7 | 46.8 |
| TD3 + BC, tuned | 78.6 (-13%) | 47.2 (-47%) | 66.1 (-27%) |
| ReBRAC w/o $\gamma$ change | - | 46.2 (-48%) | - |
| ReBRAC w/o LN | 57.3 (-36%) | 0.0 (-100%) | 52.9 (-41%) |
| ReBRAC w/o layer | 85.8 (-5%) | 17.6 (-80%) | 51.5 (-43%) |
| ReBRAC w/o actor penalty | 21.7 (-76%) | 13.9 (-84%) | 18.6 (-79%) |
| ReBRAC w/o critic penalty | 88.1 (-2%) | 62.3 (-31%) | 77.8 (-14%) |
| ReBRAC w/o large batch | 86.9 (-4%) | 60.5 (-33%) | 76.4 (-15%) |
| ReBRAC | 90.8 | 90.5 | 90.7 |

extent. Overall, disabling any single modification leads to a decline in performance on average, which highlights the essential role each change plays in achieving state-of-the-art performance.

## 5  RELATED WORK

**Ensemble-free offline RL methods.** In recent years, many offline reinforcement learning algorithms were developed. TD3 + BC (Fujimoto & Gu, 2021) represents a minimalist approach to offline RL, which incorporates a Behavioral Cloning component into the actor network loss, enabling online actor-critic algorithms to operate in an offline setting. CQL (Kumar et al., 2020) drives the critic network to assign lower values to out-of-distribution state-action pairs and higher values to in-distribution pairs. IQL (Kostrikov et al., 2021) proposes a method for learning a policy without the need for sampling out-of-distribution actions.

Despite this, in order to achieve state-of-the-art results in an ensemble-free setup, more sophisticated methods may be necessary. For instance, CNF (Akimov et al., 2022) pre-trains a normalizing flow encoder for actions, then trains the actor to predict actions in the latent space. BPPO (Zhuang et al., 2023), on the other hand, pre-trains the policy using Behavior Cloning, estimates the behavior policy's Q-function using SARSA, and executes PPO (Schulman et al., 2017) using the obtained estimations.

**Ensemble-based offline RL methods.** A significant number of works in offline reinforcement learning have also leveraged ensemble methods for uncertainty estimation. The recently introduced SAC-N (An et al., 2021) algorithm outperformed all previous approaches on the D4RL locomotion tasks; however, it necessitated large ensembles for some tasks, such as the hopper task, which required an ensemble size of 500 and imposed a significant computational burden. To mitigate this, the EDAC algorithm was introduced in the same work, which utilized ensemble diversification to reduce the ensemble size from 500 to 50. Despite the reduction, the ensemble size remains substantial compared to ensemble-free alternatives. It is worth mentioning that neither SAC-N nor EDAC are capable of solving the AntMaze tasks (Tarasov et al., 2022).

Another state-of-the-art algorithm in the locomotion tasks is RORL (Yang et al., 2022), which is a modification of SAC-N that makes the Q function more robust and smooth by perturbing state-action pairs with the use of out-of-distribution actions. RORL also requires an ensemble size of up to 20. On the other hand, MSG (Ghasemipour et al., 2022) utilizes independent targets for each ensemble member and achieves good performance on the locomotion tasks with an ensemble size of 4, but requires 64 ensemble members to achieve state-of-the-art performance on the AntMaze tasks.

**Layer Normalization.** Normalization techniques in reinforcement learning have gained some traction, despite not being as widely studied as they are in deep learning. Hiraoka et al. (2021) boosted the computation efficiency of state-of-the-art RL algorithms by adding dropout and layer normalization to critic networks. Smith et al. (2022) built upon these modifications and discovered that the majority of the improvement was due to the implementation of layer normalization. Bhatt et al. (2019); Kumar et al. (2022); Nikulin et al. (2022) also tested normalization in the offline RL

setting and showed that normalization techniques are also beneficial. A parallel study Ball et al. (2023) empirically shows that LayerNorm helps to prevent catastrophic value extrapolation for the Q function when using offline datasets in RL.

**Deeper Networks.** In natural language processing, it has been established that, with a sufficient amount of data, model performance scales with the model size (Kaplan et al., 2020). Similar trends have been observed in reinforcement learning (Sinha et al., 2020; Neumann & Gros, 2022) and offline reinforcement learning (Lee et al., 2022a; Kumar et al., 2022), however, more research is needed in this area. For instance, Fujimoto & Gu (2021) have shown that reducing the number of layers in CQL (Kumar et al., 2020) may result in a significant decline in performance. On the other hand, Zhuang et al. (2023) offer an alternative approach to increasing network size by expanding the width of the network instead of its depth.

**Large Batch Optimization.** The concept of large batch optimization (You et al., 2017; 2019) in the context of offline RL was first introduced in Nikulin et al. (2022). While this area of research holds promise, to date it has only been tested on the SAC-N Nikulin et al. (2022) and CNF Akimov et al. (2022) algorithms and has yet to be explored with other offline RL algorithms.

$\gamma$ **value change**. The discount factor $\gamma$ is well established as a critical hyperparameter in reinforcement learning (Jiang et al., 2015). A recent study Hu et al. (2022) highlights the utility of decreasing $\gamma$ for offline RL; however, the authors only consider values below $0.99$, which is the standard setting for most tasks. In our research, we have observed evidence that increasing the $\gamma$ value for AntMaze tasks may lead to improved results, and further investigation in this direction is required.

## 6 OUTLOOK

In this work, we revisit recent advancements in the offline RL field over the last two years and incorporate a modest set of improvements to a previously established minimalistic TD3 + BC approach. Our experiments demonstrate that even with these limited updates, we are able to achieve state-of-the-art performance on the D4RL benchmark for the AntMaze and locomotion tasks, or come close.

Despite the noteworthy achievements, it is imperative to explore the potential impact of other offline RL methods with the techniques we employed. Although we eagerly anticipate a positive outcome, it is also our belief that the offline RL community requires an additional benchmark, as one of the most challenging D4RL tasks, AntMaze, is almost solved.

An alternative research direction for the ReBRAC method would be to assess its performance in an offline-to-online setting. This direction appears promising due to several factors. First, the algorithm exhibits a high degree of proficiency following offline pre-training. Second, our algorithm shares similarities with TD3 + BC, which has proven effective for online fine-tuning (Beeson & Montana, 2022). Lastly, we have incorporated pessimism into the Q function, a crucial component for transitioning from offline to online environments (Lee et al., 2022b) .

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

## A  EXPERIMENTAL DETAILS

In order to generate the results presented in Table 1 and Table 2, we conducted a hyperparameter search and selected the best results from the final evaluations for each dataset. Our algorithm was implemented using JAX and the experiments were conducted on V100 and A100 GPUs.

**Locomotion tasks.** In our study, we utilized the latest version of the datasets – v2. The agents were trained for a total of one million steps and evaluated over 10 episodes.

We fine-tuned the $\beta_1$ parameter for the actor, which was selected from a range of $0.001, 0.01, 0.05, 0.1$. Similarly, the $\beta_2$ parameter for the critic was selected from a range of $0, 0.001, 0.01, 0.1, 0.5$. The selected best parameters for each dataset are reported in the appendix, in Table 6. Furthermore, the sensitivity analysis of these parameters is also presented in Appendix C.

**AntMaze tasks.** In our work, we utilized v1 of the datasets. It's worth noting that previous studies have reported results using v0 datasets, which were found to contain numerous issues[4]. Each agent was trained for 1 million steps and evaluated over 100 episodes. In accordance with Chen et al. (2022b), we modified the reward function by multiplying it by 100. We also tuned learning rate hyperparameters as it appeared to be essential for stable convergence.

The $\beta_1$ (actor) and $\beta_2$ (critic) hyperparameters were carefully selected from the respective ranges of $0.0003, 0.0005, 0.001, 0.002, 0.003$ and $0, 0.0001, 0.0005, 0.001$. In addition, the actor and critic learning rates were optimized from the ranges of $0.0001, 0.0002, 0.0003, 0.0005$ and $0.0003, 0.0005, 0.001$, respectively. The optimal hyperparameters for each dataset are presented in the appendix in Table 5.

We also modified the $\gamma$ value when addressing these tasks, driven by the following motivation. The length of the episodes in AntMaze can be as long as 1000 steps, while the reward is sparse and can only be obtained at the end of the episode. As a result, the discount for the reward with the default $\gamma$ can be as low as $0.99^{1000} = 4 \cdot 10^{-5}$, which is extremely low for signal propagation, even when multiplying the reward by 100. By increasing $\gamma$ to 0.999, the minimum discount value becomes $0.999^{1000} = 0.36$, which is more favorable for signal propagation.

## B  HYPERPARAMETERS

Table 4: ReBRAC general hyperparameters.

| Parameter | Value |
| --- | --- |
| optimizer | Adam (Kingma & Ba, 2014) |
| batch size | 1024 |
| learning rate (all networks) | 1e-3 (on locomotion, tuned for each AntMaze task) |
| tau ($\tau$) | 5e-3 |
| hidden dim (all networks) | 256 |
| num layers (all networks) | 4 |
| gamma ($\gamma$) | 0.99 (0.999 on AntMaze) |
| nonlinearity | ReLU |

Table 5: ReBRAC best hyperparameters used in D4RL AntMaze domain.

| Task Name | $\beta_1$ (actor) | $\beta_2$ (critic) | actor lr | critic lr |
| --- | --- | --- | --- | --- |
| antmaze-umaze | 0.0005 | 0.0005 | 0.0005 | 0.0005 |
| antmaze-umaze-diverse | 0.0005 | 0.0005 | 0.0005 | 0.0005 |
| antmaze-medium-play | 0.0005 | 0.0005 | 0.0005 | 0.001 |
| antmaze-medium-diverse | 0.001 | 0.0001 | 0.0005 | 0.0005 |
| antmaze-large-play | 0.002 | 0.0001 | 0.0005 | 0.0003 |
| antmaze-large-diverse | 0.003 | 0.0005 | 0.0005 | 0.0005 |

---

[4]https://github.com/Farama-Foundation/D4RL/issues/77

Table 6: ReBRAC best hyperparameters used in D4RL locomotion domain.

| Task Name | $\beta_1$ (actor) | $\beta_2$ (critic) |
|---|---|---|
| halfcheetah-medium | 0.001 | 0.01 |
| halfcheetah-medium-expert | 0.01 | 0.1 |
| halfcheetah-medium-replay | 0.001 | 0.001 |
| hopper-medium | 0.01 | 0.001 |
| hopper-medium-expert | 0.1 | 0.1 |
| hopper-medium-replay | 0.05 | 0.01 |
| walker2d-medium | 0.05 | 0.1 |
| walker2d-medium-expert | 0.01 | 0.5 |
| walker2d-medium-replay | 0.05 | 0.01 |

## C  SENSITIVITY TO HYPERPARAMETERS

Following Kurenkov & Kolesnikov (2022), we demonstrate the sensitivity of ReBRAC to the choice of $\beta_1$ and $\beta_2$ hyperparameters under uniform policy selection on locomotion tasks in Figure 1. As one can see, approximately 10 policies are required to attain ensemble-free state-of-the-art performance.

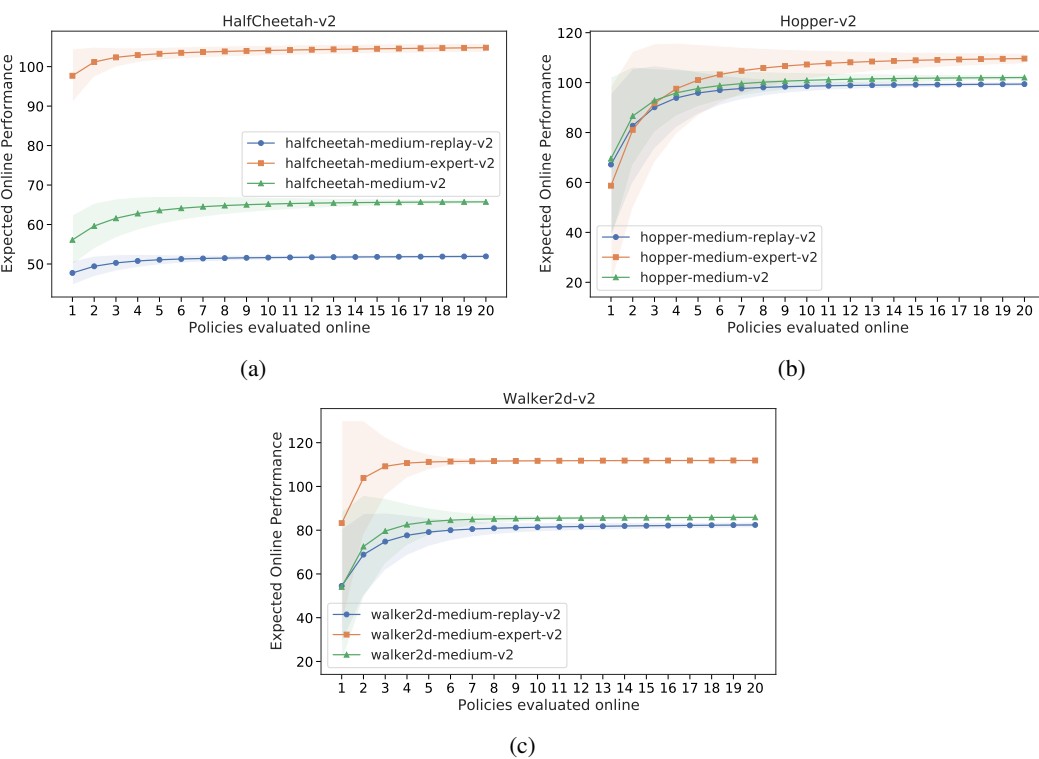

Figure 1: ReBRAC Expected Online Performance under uniform policy selection on D4RL locomotion tasks.

## D  ALL TASKS ABLATION

Table 7: ReBRAC ablations for halfcheetah tasks. We report final normalized score averaged over 4 random seeds.

| Ablation | halfcheetah-medium | halfcheetah-medium-expert | halfcheetah-medium-replay | Average |
|---|---|---|---|---|
| TD3 + BC, paper | 48.3 ± 0.3 | 90.7 ± 4.3 | 44.6±0.5 | 61.2 |
| TD3 + BC, our | 44.4 ± 0.1 | 93.5 ± 0.4 | 40.7 ± 0.8 | 59.5 |
| TD3 + BC, tuned | 59.1 ± 1.4 (-10%) | 91.3 ± 3.7 (-13%) | 49.8 ± 0.5 (-4%) | 66.7 (-10%) |
| ReBRAC w/o LN | 38.5 ± 36.9 (-41%) | 79.8 ± 10.2(-24%) | 2.7 ± 6.0 (-94%) | 40.3 (-45%) |
| ReBRAC w/o layer | 56.7 ± 10.7 (-13%) | 86.7 ± 7.3 (-17%) | 42.5 ± 3.9 (-18%) | 61.9 (-16%) |
| ReBRAC w/o actor penalty | 63.9 ± 1.3 (-3%) | 79.1 ± 13.5 (-25%) | 37.9 ± 4.3 (-27%) | 60.3 (-19%) |
| ReBRAC w/o critic penalty | 65.5 ± 4.3 (-0%) | 100.1 ± 4.5 (-5%) | 45.7 ± 4.5 (-12%) | 70.4 (-5%) |
| ReBRAC w/o large batch | 61.8 ± 0.8 (-6%) | 92.7 ± 0.9 (-12%) | 52.0 ± 3.2 (-0%) | 68.8 (-7%) |
| ReBRAC | 65.9 ± 1.1 | 105.6 ± 1.4 | 52.2±4.6 | 74.5 |

Table 8: ReBRAC ablations for hopper tasks. We report final normalized score averaged over 4 random seeds.

| Ablation | hopper-medium | hopper-medium-expert | hopper-medium-replay | Average |
|---|---|---|---|---|
| TD3 + BC, paper | 59.3 ± 4.2 | 98.0 ± 9.4 | 60.9 ± 18.8 | 72.7 |
| TD3 + BC, our | 49.2 ± 4.3 | 65.4 ± 23.6 | 42.3 ± 10.7 | 52.3 |
| TD3 + BC, tuned | 57.8 ± 4.6 (-43%) | 97.8 ± 12.7 (-11%) | 81.3 ± 22.7 (-18%) | 78.9 (-24%) |
| ReBRAC w/o LN | 9.8 ± 14.1 (-90%) | 104.0 ± 5.2 (-5%) | 76.4 ± 24.3 (-23%) | 63.4 (-39%) |
| ReBRAC w/o layer | 102.6 ± 0.7 (+0%) | 102.7 ± 9.2 (-7%) | 98.4 ± 2.3 (-1%) | 101.2 (-2%) |
| ReBRAC w/o actor penalty | 2.2 ± 1.8 (-97%) | 1.8 ± 1.3 (-98%) | 5.0 ± 4.3 (-94%) | 3.0 (-97%) |
| ReBRAC w/o critic penalty | 97.0 ± 6.4 (-5%) | 110.6 ± 1.7 (-0%) | 98.8 ± 1.6 (-1%) | 102.1 (-2%) |
| ReBRAC w/o large batch | 96.9 ± 6.4 (-5%) | 106.2 ± 3.1 (-3%) | 98.1 ± 4.1 (-1%) | 100.4 (-3%) |
| ReBRAC | 102.5 ± 0.3 | 110.6 ± 2.0 | 99.9 ± 1.1 | 104.3 |

Table 9: ReBRAC ablations for walker2d tasks. We report final normalized score averaged over 4 random seeds.

| Ablation | walker2d-medium | walker2d-medium-expert | walker2d-medium-replay | Average |
|---|---|---|---|---|
| TD3 + BC, paper | 83.7 ± 2.1 | 110.1 ± 0.5 | 81.8 ± 5.5 | 91.8 |
| TD3 + BC, our | 109.1 ± 0.4 | 74.8 ± 2.7 | 46.8 ± 17.6 | 76.9 |
| TD3 + BC, tuned | 77.7 ± 4.3 (-43%) | 110.8 ± 0.6 (-9%) | 82.9 ± 5.7 (-0%) | 90.4 (-3%) |
| ReBRAC w/o LN | 84.1 ± 2.8 (-2%) | 53.1 ± 55.7 (-52%) | 68.1 ± 7.5 (-18%) | 68.4(-27%) |
| ReBRAC w/o layer | 85.4 ± 0.5 (-0%) | 112.8 ± 0.6 (+0%) | 84.8 ± 3.2 (+1%) | 94.3(+0%) |
| ReBRAC w/o actor penalty | 1.3 ± 1.5 (-98%) | 0.1 ± 0.2 (-99%) | 4.4 ± 0.5 (-94%) | 1.9 (-97%) |
| ReBRAC w/o critic penalty | 84.8 ± 4.0 (-1%) | 111.9 ± 0.3 (+0%) | 78.8 ± 5.6 (-5%) | 91.8 (-2%) |
| ReBRAC w/o large batch | 82.6 ± 3.0 (-4%) | 111.9 ± 0.2 (0%) | 80.5 ± 6.1 (-3%) | 91.6 (-2%) |
| ReBRAC | 86.1 ± 0.2 | 111.9 ± 0.4 | 83.2 ± 10.3 | 93.7 |

Table 10: ReBRAC ablations for antmaze tasks. We report final normalized score averaged over 4 random seeds.

| Ablation | antmaze-umaze | antmaze-umaze-diverse | antmaze-medium-play | antmaze-medium-diverse | antmaze-large-play | antmaze-large-diverse | Average |
|---|---|---|---|---|---|---|---|
| TD3 + BC, paper | 78.6 | 71.4 | 10.6 | 3.0 | 0.2 | 0.0 | 27.3 |
| TD3 + BC, our | 70.2 ± 3.5 | 65.2 ± 2.8 | 0.7 ± 0.9 | 0.5 ± 1 | 0.0 ± 0.0 | 0.0 ± 0.0 | 22.7 |
| TD3 + BC, tuned | 88.0 ± 1.4 (-11%) | 79.5 ± 5.5 (-18%) | 38.5 ± 44.4 (-54%) | 28.0 ± 31.9 (-66%) | 46.2 ± 19.7 (-47%) | 3.5 ± 5.7 (-96%) | 47.2 (-47%) |
| ReBRAC w/o γ change | 82.7 ± 2.9 (-16%) | 76.5 ± 8.2 (-21%) | 37.5 ± 31.6 (-55%) | 19.5 ± 31.3 (-76%) | 13.7 ± 23.0 (-78%) | 47.5 ± 32.9 (-47%) | 46.2 (-48%) |
| ReBRAC w/o LN | 0.0 ± 0.0 (-100%) | 0.0 ± 0.0 (-100%) | 0.0 ± 0.0 (-100%) | 0.0 ± 0.0 (-100%) | 0.0 ± 0.0 (-100%) | 0.0 ± 0.0 (-100%) | 0.0 (-100%) |
| ReBRAC w/o layer | 11.7 ± 23.5 (-88%) | 94.0 ± 2.8 (-4%) | 0.0 ± 0.0 (-100%) | 0.0 ± 0.0 (-100%) | 0.0 ± 0.0 (-100%) | 0.0 ± 0.0 (-100%) | 17.6 (-80%) |
| ReBRAC w/o actor penalty | 62.5 ± 19.0 (-36%) | 18.0 ± 11.5 (-81%) | 0.5 ± 0.5 (-99%) | 0.0 ± 0.0 (-100%) | 0.0 ± 0.0 (-100%) | 2.7 ± 5.5 (-97%) | 13.9 (-84%) |
| ReBRAC w/o critic penalty | 84.7 ± 15.3 (-14%) | 97.5 ± 1 (-0%) | 36.2 ± 40.0 (-57%) | 53.5 ± 39.4 (-35%) | 81.2 ± 22.3 (-7%) | 20.7 ± 41.5 (-77%) | 62.3 (-31%) |
| ReBRAC w/o large batch | 96.2 ± 2.0 (-3%) | 95.7 ± 2.0 (-2%) | 43.5 ± 20.9 (-48%) | 38.5 ± 29.3 (-53%) | 89.5 ± 2.38 (+1%) | 0.0 ± 0.0 (-100%) | 60.5 (-33%) |
| ReBRAC | 99.2 ± 0.9 | 98.0 ± 2.1 | 84.5 ± 4.7 | 82.5 ± 7.5 | 88.2 ± 3.7 | 90.7 ± 4.8 | 90.5 |

