# OpenReview forum: "Revisiting Behavior Regularized Actor-Critic"
_ICLR.cc/2023/Workshop/RRL — RRL 2023 Poster_

### Official Review · Reviewer_mrjE · 2023-02-18
**Empirical paper on offline RL design choices. Not super related to the workshop.**

**Rating:** 2
**Confidence:** 3

**Review:**

This paper empirically studies the effect of combining deeper networks, layer normalization, and large batch size in the context of offline RL. Specifically, the paper extends BRAC algorithm with a few modifications (some borrowed from TD3+BC) that stabilize its training, and then incorporate the 3 additional modifications. The resulting ReBRAC algorithm achieves state-of-art performance on D4RL.

Strengths:
1. The paper is well-written and clear
2. The experiments are thorough; the ablations clear demonstrate the utilities of the proposed modifications

Weaknesses:
1. Hyperparameters are tuned per environment+dataset combination. In practice, you cannot do this as there is no "free" evaluation of the offline learned policies. Furthermore, this reduces the significance of the results as it is not clear whether the gains are just from better hyperparameters.

2. Some missing related works:
Sinha et al., D2RL: Deep Dense Architectures in Reinforcement Learning
Bjorck et al., Towards deeper deep reinforcement learning

Finally, I would note that this is a paper strictly in the domain of offline RL; while relevant, I do not think it is directly related to the notion of reincarnating RL.

---

### Official Review · Reviewer_Cr58 · 2023-03-01
**A good examination of design choices to minimalist offline RL**

**Rating:** 3
**Confidence:** 5

**Review:**

Summary:
The paper has two main contributions:
1. It proposes a minimalist expansion to the BRAC algorithm that uses a mean-squared error for the action deviation penalty and has the popular TD3+BC algorithm as a special case.
2. Explores design choices in the practical implementation of the algorithms, such as the use of layer normas and batch sizes.

Strengths:
The proposed algorithm is pretty simple and straightforward. The implementation details are useful and seem to make a significant difference in the overall results.


Weaknesses:
The paper evaluates it’s results on the standard d4rl benchmark locomotion tasks. With so many papers, approaches, design choices and hyper-parameters fine -tuning (from the paper: “β1 and β2 parameters from Equations 3 and 4 are carefully tuned”)  it is not clear whether the results are indeed significant or a matter of over-fitting to the datasets. It would be helpful to evaluate the more complex domains of the benchmark as well. This is also a completely offline single-task RL approach, which might not be the best fit for the workshop theme.


Conclusion:
I would still recommend and accept as the proposed approach is simple and still achieves good results. The design experiments are useful to practical implementations and there is a renewed interest in this recently.